# Smart Shelf System for Customer Behavior Tracking in Supermarkets

**DOI:** 10.3390/s24020367

**Published:** 2024-01-08

**Authors:** John Anthony C. Jose, Christopher John B. Bertumen, Marianne Therese C. Roque, Allan Emmanuel B. Umali, Jillian Clara T. Villanueva, Richard Josiah TanAi, Edwin Sybingco, Jayne San Juan, Erwin Carlo Gonzales

**Affiliations:** 1Department of Electronics and Computer Engineering, Gokongwei College of Engineering, De La Salle University, 2401 Taft Avenue, Malate, Manila 1004, Metro Manila, Philippines; christopher_bertumen@dlsu.edu.ph (C.J.B.B.); marianne_roque@dlsu.edu.ph (M.T.C.R.); allan_umali@dlsu.edu.ph (A.E.B.U.); jillian_villanueva@dlsu.edu.ph (J.C.T.V.); edwin.sybingco@dlsu.edu.ph (E.S.); 2Department of Manufacturing Engineering and Management, Gokongwei College of Engineering, De La Salle University, 2401 Taft Avenue, Malate, Manila 1004, Metro Manila, Philippines; richard.tanai@dlsu.edu.ph; 3Department of Industrial and Systems Engineering, Gokongwei College of Engineering, De La Salle University, 2401 Taft Avenue, Malate, Manila 1004, Metro Manila, Philippines; jayne.sanjuan@dlsu.edu.ph; 4Management and Organization Department, Ramon V. del Rosario College of Business, De La Salle University, 2401 Taft Avenue, Malate, Manila 1004, Metro Manila, Philippines; erwin.carlo.gonzales@dlsu.edu.ph

**Keywords:** smart shelves, visual analytics, retail analytics, computer vision, sensor fusion

## Abstract

Transactional data from point-of-sales systems may not consider customer behavior before purchasing decisions are finalized. A smart shelf system would be able to provide additional data for retail analytics. In previous works, the conventional approach has involved customers standing directly in front of products on a shelf. Data from instances where customers deviated from this convention, referred to as “cross-location”, were typically omitted. However, recognizing instances of cross-location is crucial when contextualizing multi-person and multi-product tracking for real-world scenarios. The monitoring of product association with customer keypoints through RANSAC modeling and particle filtering (PACK-RMPF) is a system that addresses cross-location, consisting of twelve load cell pairs for product tracking and a single camera for customer tracking. In this study, the time series vision data underwent further processing with R-CNN and StrongSORT. An NTP server enabled the synchronization of timestamps between the weight and vision subsystems. Multiple particle filtering predicted the trajectory of each customer’s centroid and wrist keypoints relative to the location of each product. RANSAC modeling was implemented on the particles to associate a customer with each event. Comparing system-generated customer–product interaction history with the shopping lists given to each participant, the system had a general average recall rate of 76.33% and 79% for cross-location instances over five runs.

## 1. Introduction

Retail analytics involves the analysis of data to provide insight into different aspects of retail operations. Through this type of analytics, trends in sales and customer behavior may be characterized [1]. Retailers are able to make data-driven decisions regarding business strategies related to pricing, product placement, and more. These strategies may help reduce costs, increase sales, and improve the overall customer experience [2]. When it comes to inventory management, neither a surplus nor deficit are beneficial. Ideally, the supply should sufficiently meet customer demand to prevent products remaining in inventory for prolonged periods of time which may lead to deterioration or eventual discarding, which would incur losses [3]. Hence, inventory management is one of the most crucial considerations when it comes to data analytics in the retail industry.

Point-of-sale (POS) systems monitor and execute transactions, making them the most common data source for retail analytics in physical establishments. Although this system effectively tracks sale volumes, more data is required to accurately provide information regarding product sales and availability relative to customer response [4]. This is also taking into consideration that additional POS system features may be kept behind a paywall that puts micro, small, and medium enterprises (MSMEs) at a disadvantage compared to their well-established competitors, who have more resources [5]. Additionally, unlike online retailers, which can monitor non-purchasing customer behavior, final purchase information from onsite retail stores is limited to data provided by POS systems [6].

Further research suggests that implementing smart shelf systems is an effective way of obtaining additional and unseen data from existing methods in physical retail stores. Several approaches have been made in designing such systems, all of which generally focus on customer interaction with products [4,7,8,9]. An autonomous shopping solutions provider [10] uses a scalable multi-camera approach to account for problems with occlusion. However, the processing of vision data is computationally intensive. Another approach for implementing smart shelves is making use of cameras and weight sensors [11]. However, its main limitation is the degraded system performance, due to the system not being able to detect cross-location or a situation when a person reaches for a product in different weight bins.

We take the view that the difficulty of cross-location [11] is rooted in the lack of object identifiers tracked in a video across a time period, known as track ID, from a multi-object tracker. Furthermore, a multi-product multi-person setting further leads to a noisy trajectory for track ID. Our study aims to develop a smart shelf system incorporating PACK-RMPF that solves cross-location problems while alleviating the effect of a noisy trajectory. Enhanced by this approach, our smart shelf systems evaluate customer interactions with fixed, shelved products through a combination of sensors and object recognition.

The main contributions of this study are as follows:A smart shelf system that uses single camera and weight sensor arrays for a multi-person, multi-product tracking systemA PACK-RMPF sensor fusion algorithm that enhances the tracking of product interaction by associating weight event data and multi-object tracking data with the use of state filters and inlier estimators to address occlusion and localization problems and cross-location.A simulated supermarket experiment setup that evaluates system performance based on a customer’s journey of purchasing goods based on a pre-defined shopping list.

## 2. Related Work

Generally, inventory management seeks to maximize the collection and use of inventory-related data. In the case of retail stores, it often involves streamlining an inventory with consideration for customer demand and carrying costs, among other factors and risks. One aspect of inventory management is the tracking of goods as they move from an inventory to customers. The emergence of smart retailing, through various technologies that enable more advanced retail analytics, aims to extend the functionality of POS systems from merely tracking inventory movement to making logical inferences about how customer behavior and customer–product interactions are correlated with the turnover of products in an inventory. Smart shelves, a common approach to smart retailing, utilize a variety of ways to extract pertinent data that can be used to predict inventory with respect to customer behavior, including, but not limited to, weight sensors, RFID tags, and specialized cameras [12].

Higa and Iwamoto [13] proposed a low-cost solution involving surveillance cameras as a tool for tracking the amount of products on a shelf in 2019. The method utilized was background subtraction wherein the background of an image is removed before the foreground—which contains the products—is observed. Through the use of a CNN, the system compares fully replenished, partially stocked, and empty shelf conditions. At the end of the research, the system was able to garner an 89.6% accuracy rate. The authors proposed that this data could eventually be utilized to improve profits in retail stores by improving the shelf availability of products [13]. A similar solution was proposed back in 2015 [14]. In this solution, it was suggested that image detection and processing should alert store managers upon the need to replenish shelves, as well as the discrepancies detected such as misplaced products.

When it comes to the use of weight sensors for product tracking, physical interactions with a product have been typically tracked through changes in weight sensor data, as determined by algorithms. Ref. [8] utilized a bin system in their research wherein a single shelf row was divided into bins that had dedicated load cells for tracking a specified type of product instead of having a single platform for all available products on the row. In [15], each shelf row was equipped with load cell weight sensors arranged so that positional tracking would be possible on the shelf. This approach worked through constant weight readings from the load cells and an algorithm for detecting significant changes in the weight of a load cell in the system to be registered as a pick up. Through the use of support vector regression (SVR) and artificial neural networks (ANNs), the position of the product could also be predicted.

Unmanned retail stores, otherwise referred to as cashier-less stores, utilizing smart shelf systems typically implement human pose estimation (HPE) for customer tracking. Implementing HPE for customer tracking is advantageous—especially in solving action recognition problems—due to the plethora of pre-trained data readily available to use [16]. As an example, [17] developed a system for an unmanned retail store that makes use of visual analytics and grating sensors. With the grating sensors emitting infrared beams, the creation of an invisible curtain-like barrier was formed. Once a customer’s hand passed through the curtain, the infrared beams would be interrupted and the motion would be flagged as an item being taken. One camera was integrated with a human pose estimation algorithm to classify the hand action when such an event occurred. The rest of the cameras that were installed were used to determine which product was taken. Mask R-CNN was utilized for multiple customer tracking. Under this model, the body was tracked as one silhouette. Furthermore, under this system, occlusions were solved by adding the possible losses in each part of the silhouette.

Smart shelf systems vary in terms of the extent of weight sensor and computer vision integration. For those that primarily or solely use computer vision, there is usually a focus on security, autonomous or cashier-less checkouts, or visual indicators of customer behavior. AiFi, asserting itself as a provider of autonomous shopping solutions, uses an extensive amount of cameras to be able to effectively run stores [10]. In addition to occlusion problems, there may be a question of redundancy depending on the actual system design. The in-store autonomous checkout system (ISACS) for retail proposed by [11], on the other hand, was divided into three major tracking subsystems: product tracking with weight sensors, customer tracking through cameras whose feeds underwent human pose estimation, and multi-human to multi-product tracking through sensor fusions. These systems were highly dependent on the pose estimation in associating the weight event detected with the person. However, a major limitation of these systems is the problem of collision and cross-location, especially when multiple customers are interacting with a shelf.

Hence, there are several approaches for implementing sensor fusions. An example of this approach is making use of a particle filter, which can be divided into four stages: generation, prediction, updating, and resampling. Particles are uniformly generated, and the positions of the generated particles are predicted. The sensor readings are compared with the predicted position of each particle, and the probability that relates the distance of the particles to the measured position from a sensor is calculated. The particles are then resampled so that only the nearer particles are measured. Finally, the algorithm can retrieve the trajectory of a moving object. We were able to recreate the trajectory of the wireless signals, sourcing from the moving objects, through the particle filter, and we recommend this as a solution for localization problems in sensor fusion.

## 3. Smart Shelf System Design

### 3.1. Shelf Physical Overview

The shelf tested was a double-sided gondola shelf equipped with two rows, 0.5 m apart, on each side. Each row contained three (3) weighing platforms primarily operational through a pair of straight bar load cells, hereinafter referred to as weight bins, for a total of twelve (12) weight bins. Distinct products were placed on each weight bin, corresponding to twelve (12) products. The number of units per product varied based on the dimensions of each product. The camera was located 2.43 m away from the shelf and 2.26 m from the ground, as depicted in Figure 1.

### 3.2. System Overview

PACK-RMPF utilizes the data from the weight sensors and video feed per unit time. For weight sensor data, the signals undergo a filter process as a safeguard against extreme signal spikes, movements, or errors. Extreme signal spikes and movements are referred to weight signal changes where the weight signal steps are deemed impossible. For instance, a 300 g product is deemed impossible to move higher than 1 kg of a weight signal step considering that the products are removed one at a time. Additionally, errors in retrieving weight values will send out an error string instead of float values. Thus, for extreme signal spikes, movements, or errors, forward data filling using the padding method was utilized. Afterward, the signal underwent a weight event detection algorithm. A three-period moving average and a three-period moving variance were performed. Each product contained a specific moving variance threshold based on the initial calibration results for each product. Thus, the start of an event was triggered when the moving variance was higher than the threshold two consecutive times, while the end of its event was triggered when the moving variance was lower than the threshold two consecutive times. The moving average was utilized to handle possible volatility. As such, it was utilized for determining the weight value at the start and end of an event. When a weight sensor decreased by the end of an event, it was labelled a pick up; likewise, if it increased by the end of an event, it was labelled a putback.

The data for the vision system contained the detected customers per frame attached to a unique track ID, a bounding box, and keypoint detections. Since the vision system had Re-ID features, it was able to trace a lost tracked bounding box, typically due to occlusion, back to the customer using their appearance features [18]. It was equally important that only the customer’s hand were extracted from the human pose estimation (HPE), since it was the only one of significance to the system.

The integration system used a particle filter for trajectory tracking of the bounding box and the hand keypoints. These trajectories were processed via random sample consensus (RANSAC) to identify the correct customer who performed the weight event. With that, the data process of PACK-RMPF is summarized in Figure 2. 

At the end of the integration system where the product and the customer’s behavior are associated, PACK-RMPF is able to record each customer’s action. Likewise, the system is able to process, derive, and obtain the history of each customer’s action, the current products in their cart, the current inventory status of the shelf, the average retention time of each product, and the number of specific actions the customer performed.

## 4. Weight Change Event Detection System

### 4.1. Weight Bin

Each weight bin consisted of a pair of straight bar load cells. An acrylic sheet was mounted on top of the load cells to aid in weight distribution. The load cells were also screwed onto the shelf. Each load cell had a capacity of 3 kg for a total capacity of 6 kg per weight bin. The dimensions of each weight bin are listed on Figure 3. 

As illustrated in Figure 4, the wires of each load cell were connected in parallel when interfaced with the inputs of an HX711 amplifier module. By default, the module had a gain of 124 and a corresponding sampling rate of 80 Hz. Each module was connected to a Raspberry Pi 4B that would then process the readings from each weight bin concurrently. The ground, supply, and clock pins for each HX711 module were the same, while each digital output was connected to a dedicated GPIO pin.

### 4.2. Weight Sensor Array Hardware

The weight sensor hardware system is shown in Figure 5. Every weight bin has a unique assignment on the physical location along the double-sided shelf. Furthermore, each weight bin is assigned a unique product item. Each row on one side of the shelf is partitioned into three slots, leading to three weight bins per row. On each side of the shelf, there are two rows, a top row and a bottom row. A total of 18 weight bins are connected to Raspberry Pi 4B, which acts as the controller.

The controller was interfaced with a host computer using secure shell (SSH) protocol. The host computer also served as the network time protocol (NTP) server to set the weight reading timestamps. Through the network, a server would be initialized to enable the RPi to continuously send timestamps and weight readings for each weight bin to a client device.

### 4.3. Weight Change Event Detection System

An overview of the weight sensor data processing, with the main goal of identifying and classifying weight change events, is found in Figure 6. The weight sensor data acquired from the controller are grouped by specific weight bin location λ. Forward filling is also implemented so that invalid readings from a weight bin assume the value of the last valid reading from the same weight bin. Then, processing is conducted based on three-period moving variance values.

The moving mean $\mu_{t}$ and moving variance $\nu_{t}$ calculate the mean and variance along a specific windowing period, respectively. N is the window length and $w_{t}$ is the weight values at time t. We define the ff as follows:(1)$$\mu_{t}=\frac{1}{N}\sum_{i=t-N}^{t}w_{i}$$
(2)$$\nu_{t}=\frac{1}{N}\sum_{i=t-N}^{t}{\left(w_{i}-\mu_{t}\right)}^{2}$$

Each weight data stream is continuously processed via its moving variance, $v_{t}^{\lambda }$. The weight change event detection, $\Delta \nu_{t}^{\lambda }$, is shown in Equation (3). It is considered to be detected once a weight data stream, $v_{t}^{\lambda }$, goes above a pre-defined product threshold, $T_{p}^{\lambda }$.
(3)$$\Delta \nu_{t}^{\lambda }=\left\{\begin{array}{cc} 1 & ,\;v_{t}^{\lambda }\geq T_{p}^{\lambda }\\ 0 & ,\;v_{t}^{\lambda }<T_{p}^{\lambda } \end{array}\right.$$

The product threshold $T_{p}^{\lambda }\;$is calibrated based on the product weight per unit and empirical repeated testing conducted with the corresponding weight bin. Table 1 summarizes the thresholds used for each weight bin.

Once a weight change event is detected, it must be further classified as either a customer picking up a product (pick up) or a customer putting back a product (put back).

When the moving variance threshold is surpassed, the algorithm considers the specific weight event involved as a significant weight change that must be classified as a pick up or a put back. With this, a sliding window mechanism is implemented to determine two consecutive events. The start of an event for each product is determined when the moving variance exceeds the moving variance threshold two consecutive times. Likewise, the end of an event for each product is determined when it is below the moving variance threshold two consecutive times. The average readings of the start and the end of an event are then compared. A weight change event would be flagged as a pick up if the reading at the end of the event was less than that at the start of the weight event. On the other hand, a weight event would be flagged as a put back if the reading at the end of the event was greater than that at the start of the weight event.

## 5. Vision System

### 5.1. Physical Camera

In this study, a webcam was considered. The A4Tech PK910P has a resolution of 720p and served as the primary camera during the testing of the complete, integrated system. Moreover, the live processing of vision data was simplified with this camera. To facilitate this processing, a computer equipped with an Intel Core i7 CPU and a dedicated NVIDIA GeForce RTX 3060 GPU was used.

### 5.2. Extraction of Multi-Object Trajectory

Primarily, the vision system aims to track and assign an identifier to customers interacting with the shelf. We present the vision system in Figure 7. This process is handled by Keypoint R-CNN [19] and StrongSORT [18].

The Keypoint R-CNN used was the built-in PyTorch model (accessed on 7 November 2023 https://pytorch.org/vision/main/models/keypoint_rcnn.html) with ResNet-50 [20] and feature pyramid networks [21] as the backbone. Our Keypoint R-CNN used a pre-trained model from the COCO dataset [22]. Each person identified via the model was assigned to have 17 keypoints. Alongside with these keypoints, the corresponding bounding box was also detected via the model. Only those detections having a confidence score of 80% or higher were considered—this rule was put in place to filter out false positive detections. Each bounding box was assigned an ID through StrongSORT, which is an MOT baseline that utilizes a tracking-by-detection paradigm approach. This tracked and associated objects in a scene through the appearance and velocity of the objects. The most pertinent keypoints from Keypoint RCNN were keypoints 9 and 10 which tracked the left and right hands, respectively. After performing both vision processes, the bounding box information and ID, keypoints 9 and 10, and the timestamps for each frame were exported to a comma separated- values (CSV) file.

Furthermore, the vision system utilized a pre-trained StrongSORT model for assigning each customer a track ID and for Re-ID purposes. Its detector was trained on the CrowdHuman dataset [20,21] and the MOT17 half-training set [20]. The training data were generated by cutting annotated trajectories, not tracklets, with random spatiotemporal noise at a 1:3 ratio of positive and negative samples [20]. Adam was utilized as the optimizer, and cross-entropy loss was utilized as the objective function and was trained for 20 epochs with a cosine annealing learning rate schedule [20]. For the appearance branch, the model was pretrained on the DukeMTMC-reID dataset [20]. In the study, strongSORT Re-ID weights utilized osnet_x_25_msmt17. Additionally, strongSORT was equipped with human pose estimation using a pre-trained Keypoint RCNN, Resnet 50 fpn model. Its purpose was to detect the hands of the customers, which would later be utilized in the integration system to track the trajectories of a customer’s hand.

## 6. PACK-RMPF

### 6.1. PACK-RMPF Initialization

The fusion of the weight detection system data with the vision data system is performed via the product association with customer keypoints (PACK) through RANSAC modelling and particle filtering (RMPF). This includes the implementation of a particle filter and random sample consensus (RANSAC). For the particle filter, objects are tracked with respect to a landmark. Specifically, the landmarks were placed at the center of each weight bin based on its location in a frame. A sample of the points of each landmark is provided in Figure 8. 

A pre-defined rectangular area is created for each weight bin, as shown in Figure 9, for which a RANSAC model of all possible locations of the keypoints bounded by the box is created.

### 6.2. PACK-RMPF System and Cross-Location

The PACK-RMPF algorithm associates the customers with the interacted product using the multiple particle filter and RANSAC modeling. The timestamps from the weight and vision data are matched for the purpose of synchronization. Each detected customer’s keypoints are filtered in the multiple particle filter where the particles are compared to the RANSAC model. Each detected customer outputs an inlier score. The customer with the highest inlier score is associated with the interacted item. Each block of the PACK-RMPF algorithm shall be discussed in detail in the following sections. Furthermore, a summary of all the blocks is provided in Figure 10. 

#### 6.2.1. Timestamp Matching

As shown in Algorithm 1, the timestamp matching algorithm aims to match a weight event action $w_{t}$*,* through its weight event timestamp t, to vision attributes $V_{\tau }\;$and its event timestamp τ. The vision attributes include the track ID information with its corresponding bounding box and left and right hand keypoint coordinates. The algorithm iterates based on the length of the weight event actions $w_{t}$. The timestamp matching algorithm begins by calculating the Δtime of $w_{t}$. The following is the formula for Δtime*:*(4)Δtime=t−3.5 seconds

Then, the algorithm tries to find where the timestamps of t and τ are equal—or if there are no equal timestamps—within the 500 ms latency. Then, from Equation (1), all vision attributes associated with a vision timestamp $V_{t}$ within the range of the Δtime are selected. The selected $V_{t}$ are stored alongside τ,  t, and $w_{t}$ for processing in the multiple particle filter block.
**Algorithm 1:** Timestamp Matching Algorithm1:**Input:**weight event timestamp (t$),\;\mathrm{weight}\;\mathrm{event}\;\mathrm{action}\;(w_{t}$), vision event timestamp (τ$),\;\mathrm{vision}\;\mathrm{Attributes}\;(V_{\tau })$2:**Output**$:\;\mathrm{matched}\;\mathrm{weight}\;\mathrm{event}\;\mathrm{timestamp}\;\mathrm{with}\;\mathrm{vision}\;\mathrm{event}\;\mathrm{timestamp}\;(M_{\tau }$)3:**For Each** iterations conducted 4:          Calculations are made as follows: 5:          Find where t = τ or τ$\;\mathrm{is}\;\mathrm{within}\;500\;\mathrm{ms}\;\mathrm{of}\;w_{t}$6:          $\mathrm{Store}\;w_{t}$$\;\mathrm{and}\;\mathrm{selected}\;\mathrm{vision}\;\mathrm{attributes}\;\mathrm{in}\;\mathrm{one}\;M_{\tau }$ array7:          $\mathrm{Store}\;\;t,{\;w}_{t}$$,\;\mathrm{and}\;\mathrm{selected}\;\mathrm{vision}\;\mathrm{attributes}\;\mathrm{in}\;\mathrm{one}\;M_{\tau }\;$array8:**End**

#### 6.2.2. Multiple Particle Filter

The purpose of the multiple particle filter is to be able to track the trajectory of the bounding box and hand keypoints. The knowledge of the trajectory of the system is what enables PACK-RMPF to be able to determine events where cross-location occurred. Algorithm 2 describes the system flow of the particle filter.
**Algorithm 2:** Multiple Particle Filter Algorithm1:**Input:** $\left(M_{\tau }\right)$$\;\mathrm{containing}\;\mathrm{matched}\;\mathrm{vision}\;\mathrm{attributes}\;(V_{\tau }$$):\;\mathrm{customer}\;\mathrm{track}\;\mathrm{ID}\;(V_{\tau T\_ID}$$),\;\mathrm{bounding}\;\mathrm{box}\;\mathrm{coordinates}\;(V_{\tau bbox}$$),\;\mathrm{Keypoint}\;9\;\mathrm{coordinates}\;(V_{\tau K9}$$),\;\mathrm{Keypoint}\;10\;\mathrm{coordinates}\;(V_{\tau K10}$) 2:**Output**$:\;\mathrm{Bounding}\;\mathrm{box}\;\mathrm{particles}\;(P_{bbox}$$),\;\mathrm{Keypoint}\;9\;\mathrm{particles}\;(P_{K9}$$),\;\mathrm{Keypoint}\;10\;\mathrm{particles}\;(P_{K10}$$)\;\mathrm{of}\;\mathrm{each}\;\mathrm{track}\;\mathrm{ID}\;(V_{\tau T\_ID}$)3:**For** $\mathrm{len}(M_{\tau }$) iterations, follow the following procedure:4:
    $\mathrm{Extract}\;\mathrm{the}\;\mathrm{vision}\;\mathrm{attributes}\;\mathrm{inside}\;M_{\tau }$
5:    **If** ${V_{\tau }}^{1}$, **then**6:        Initialize random particles P$\;\mathrm{with}\;\mathrm{coordinates}\;(P_{x},\;P_{y}$) based on the $\mathrm{initial}\;\mathrm{coordinates}\;\mathrm{of}\;{V_{\tau }}^{1}$7:
        $\mathrm{Compute}\;\mathrm{the}\;\mathrm{distance}\;\mathrm{between}\;\mathrm{the}\;\mathrm{landmark}\;\mathrm{and}\;\mathrm{initial}\;\mathrm{coordinates}\;{V_{\tau }}^{1}$
8:        Move the particles diagonally by adding a diagonal distance to the particles 9:        Calculate for the weights of each particle10:        Apply systemic resampling11:        Store particles (P) to global particles repository 12:
    **Otherwise**
13:            Extract global particles (P) $\mathrm{containing}\;\mathrm{current}\;P_{bbox}$${,}_{\;}P_{K9}$$,\;P_{K10}$14:            Compute the distance between the landmark and the current $\mathrm{coordinates}\;{V_{\tau }}^{n}$
15:            Move the particles diagonally by adding a diagonal distance to the particles 16:            Calculate the weights of each particle17:            Apply systemic resampling18:            Store particles (*P*) to global particles repository 19:**End**20:Collect final particles (P) $\mathrm{containing}\;\mathrm{final}\;P_{bbox}$${,}_{\;}P_{K9}$$,\;P_{K10}$

The particle filter process can be summarized into three steps. These steps are predict, update and resample. The predict step is used to predict the trajectory of the keypoints [22]. For the context of PACK-RMPF, the predict step begins by generating uniform random particles based on the initial position of the vision attributes, $V_{\tau }$, consisting of the coordinates of the position of the bounding box and the hands. Then, each particle is moved, with one unit moved to the x axis and one unit moved to the y axis.

Then, the particles are updated based on their position relative to the weight bin landmarks. The following equation was utilized to calculate the position of the particles with respect to the landmarks [22]:(5)$$distance=\sqrt{{\left(V_{Ax}-x_{l}\right)}^{2}+{\left(V_{Ay}-y_{l}\right)}^{2}}$$
where $V_{Ax}$ and $V_{Ay}$ are the coordinates of the vision attributes, and $x_{l}$ and $y_{l}$ are the coordinates of the weight bin landmarks.

Finally, the particles are resampled. The following principle of systematic resampling was utilized, where ${\;w}_{k}\;$ represents the weights based on the distance of each particle, and $N_{s}$ is the number of particles [22]:(6)$$N_{eff}=\frac{1}{\sum_{i=1}^{N_{S}}{\left(w_{k}^{i}\right)}^{2}}$$

After processing the associated vision attributes to the weight event, the final particles are then evaluated in RANSAC.

#### 6.2.3. Random Sample Consensus (RANSAC)

RANSAC or random sample consensus associates the trajectory produced by the multiple particle filter with the right person who interacted with the smart shelf. RANSAC in this study utilized the SciPy library. This library created a RANSAC model through the use of the possible points inside each weight bin. The model utilized the mean absolute deviation of the possible points of the weight bin to create the model. The following is the formula utilized for the mean average deviation [23]:(7)$$MAD=\frac{\sum \left|y_{bin}-\mu_{ybin}\right|}{N_{ybin}}$$
where $y_{bin}$ is the possible y coordinates in the weight bin; $\mu_{ybin}$ is the mean of these possible y coordinates; and $N_{ybin}$ is the number of possible y coordinates that could land in the weight bin. The algorithm creating the RANSAC Weight Bin is illustrated in Algorithm 3.
**Algorithm 3:** RANSAC Weight Bin Training1:**Input:** The top left and bottom right coordinates of the pre-defined rectangular area 2:**Output**: RANSAC model of the weight bin3:Generate all possible keypoints based on the top left and bottom right coordinates of the pre-defined rectangular area4:Calculate the mean absolute deviation of the possible keypoints5:Generate the RANSAC linear regression model 

After generating the RANSAC model, the particles from the multiple particle filter are evaluated. This process is illustrated in Algorithm 4.
**Algorithm 4:** RANSAC evaluation for each trajectory1:**Input:**states S $\mathrm{containing}\;S_{centroid}$$,\;S_{K9}$$,\;S_{K10}$2:**Output**: number of inliers I3:**For Each** $s\;\mathrm{in}\;(S_{centroid}$$,\;S_{K9}$$,\;S_{K10}$),4:
         X
, Y=s
5:      ${\;y\;}_{pred}$= RANSAC weight bin model (X)6:
        $A_{dev}=|y_{pred}-y|$
7:
        **If** 
$A_{dev}<MAD$
8:
               classify as inlier I
9:
        **Otherwise**
10:
               classify as outlier O
11:**End**12:**Return** number of inliers13:Trajectory with the highest number of inliers is associated to the weight event

The trajectories of the three states S*,* centroid ($S_{centroid})\;$ and the left ($S_{K9}$), and right ($S_{K10}$) hands, are evaluated. The first step of the RANSAC evaluation involves feeding the x-coordinate of S to the generated RANSAC weight bin model. The predicted y-coordinate is evaluated based on its absolute deviation. The absolute deviation is as defined as follows:(8)$$A_{dev}=|y_{predicted}-y_{model}|$$
where $y_{predicted}$ is the predicted y value from the states, and $y_{model}$ is the y-coordinate from the RANSAC weight bin model.

After computing, if the $A_{dev}$ is less than MAD, then it is tagged as an inlier. However, if the $A_{dev}$ is greater than MAD, then it is classified as an outlier. Finally, the trajectory with the highest count of inliers is the one associated with the event. Finally, the collated associated weight events with the trajectories are exported as a CSV File.

## 7. Results and Discussions

### 7.1. Simulated Supermarket Experimental Setup

The system performance of PACK-RMPF was evaluated by implementing a simulated supermarket setup. A simulated supermarket setup is a setup that emulates a supermarket. In this setup, participants were given a list of instructions detailing which products to pick up and put back and from which part of the shelf. Furthermore, there were instances where customers deviated from what was given in the list. These deviations from the lists were recorded by the proponents. Moreover, items with cross-location were also mentioned in these lists. Overall, a total of forty-four (44) shopping lists were distributed to all participants. Videos were stored in secure file storage accessible only to the authors. No personally identifiable information, such as faces, were collected from the video recordings.

The simulated supermarket setup was implemented by dividing the duration of each test to four 30 min tests and a single 60 min test. During these durations, customers were expected to interact with the shelf according to their shopping lists. The simulated supermarket setup was implemented in one of the research laboratories of De La Salle University where the demographics of the participants were students. The duration of the experimentation lasted two days. The first day was dedicated to the 30 min tests, while the second day was dedicated to the 60 min test.

From these tests, the system performance was evaluated based on its recall rate, as defined in the equation below [11]:(9)$$\%\;Correct\;Association=\frac{Identified\;Correct\;Items}{Identified\;Correct\;Items+Unidentified\;Correct\;Items}$$

PACK-RMPF tracks the bounding box and the hands of each customer. That is why, to further check the system performance of the system, the proponents tested the components of PACK-RPMF separately.

### 7.2. Weight Event Detection System Results

The weight event detection system is designed to identify pickups and putbacks. This is necessary in order to kickstart the process of associating customer interactions with customers. All in all, the weight event detection system successfully identified pick up and put back events that happened in quick succession of each other. A single event included a minimum of two points of data, equivalent to two seconds, given the rate that weight readings were acquired. In this study, the quick weight change events evaluated spanned as long as four rows of the time series data equivalent to four seconds. It may have been possible to have more responsive weight event detection if the frequency of the weight sensor readings had been increased. This process is illustrated in Figure 11.

For each weight bin, there were minimal false or error weight readings. Weight Bin 7, corresponding to the platform with water, exhibited the most errors at 0.10% with 10 false readings among 9,753 data points. On average, 0.03% of the readings per weight bin returned were false. Evaluating each run, as low as 0.005866% and not more than 0.01725% of readings returned false. The overall performance report of each weight bin is found in Table 2.

### 7.3. Vision System Results

The vision system plays an important role in identifying the customers interacting with the shelf. The vision system assigns each customer a track ID. The track ID serves as the identifier and is used to associate a weight event with a customer.

Figure 12 illustrates that the vision system could reliably track up to three customers at a time roaming around the double-sided shelf. It was possible for the system to detect and track upward of five people at a time, even without pre-training the StrongSORT model. However, adjustments to StrongSORT parameters, a different camera angle, or multiple cameras may still be needed to improve reliability.

Based on these results, the developed smart shelf system reliably accommodates up to two customers at a time. It must be noted that the reliability was greater in instances where only one customer at a time was present on each side of the shelf. In cases where adjacent occlusion was minimal or did not occur around the time that weight events occurred, up to three customers could be detected, tracked, and associated with some reliability.

### 7.4. PACK-RMPF Simulated Supermarket Setup Results

The simulated supermarket setup aims to check the system performance of PACK-RMPF. PACK-RMPF tracks the bounding box and the hands of each customer. These keypoints play a big role in associating product movement with the customers in the frame. Furthermore, tracking these keypoints is what enables PACK-RMPF to associate a product with a customer during scenarios with cross-location. Hence, the proponents performed experimentation and tried gauging the performance of PACK-RMPF if PACK-RMPF only tracked (1) the bounding box, (2) the hands, and (3) both the bounding box and the hands of each customers.

The previous literature has added emphasis on tracking head keypoint [11]. However, PACK-RMPF gives weight on tracking the centroid of the bounding box. Table 3 illustrates the average percent of associations per run when PACK-RMPF is calibrated to track only the bounding box of each customer.

The iteration of each run consisted of scenarios where (1) there were multiple participants interacting with the shelf and (2) there was a single participant interacting with the shelf. Overall, PACK-RMPF, when tracking only the bounding box of the customer, yields an overall average percent association of 78.67%.

In the previous literature, tracking only the head has worked well when there was only one customer interacting with the shelf. The accuracy could reach as high as 100% [11]. PACK-RMPF when tracking the bounding box also reaches this same accuracy in single customer scenarios. However, incorrect associations increase as the number of participants interacting with the shelf increases. In the case of PACK-RMPF, the overall percent of association—only considering the scenarios with multiple customers—yields an average percent of association of around 57%.

The density of people interacting with the shelf affects the system performance of smart shelves due to occlusions [11]. Similarly, this was the case with PACK-RMPF. Correct associations considered whether the expected weight events triggered by a customer were associated with that customer. Upon analyzing the integrated data and reviewing the camera footage, false associations were mainly attributed to instances of adjacent occlusions—where customers were occluded from the camera’s point of view while another customer is on the same side of the shelf—due to customer stalling or customers entering in groups.

In the example shown in Figure 13, the keypoints of the occluded customer were able to be determined by the system. However, the tracked bounding box was not assigned since the confidence score threshold was not met. Although there were instances where the customer was not occluded, factors such as the 3.5 s interval considered by the integration system typically led to a higher RANSAC inlier score for the non-occluded customer standing closer to the camera.

Similarly, the same trend, as illustrated in Table 4, can also be seen with PACK-RMPF that only tracks the hands of the participants.

The hands approach of tracking in PACK-RMPF garnered an almost similar 78% overall average percent of association. But, what sets the hands approach of PACK-RMPF apart is the fact that it was able to garner an average percent of association of 58% in the multiple people tracking scenario. Even if the hands were occluded from the shelf, the system was still able to garner a higher score compared to previous approaches that only garnered 40% accuracy [11]. Similarly, the probable source of inaccuracies in associating the customer with the product is adjacent occlusion which was discussed in the previous paragraph.

Finally, Table 5 illustrates the results of using PACK-RMPF as a whole system, which combines both hand tracking and bounding box tracking, achieved an average of 76.33% correct associations of customers with weight events per run.

### 7.5. PACK-RMPF with Cross-Location Results

A persistent limitation of threshold-based approaches in related studies that sought to associate customers with weight sensor data is the instance wherein customers interact with products that they were not standing directly in front of [11]. This includes the picking up or putting back of products on the same side of the shelf. Figure 14 provides a visual example of an instance of a cross-location pick up. 

In this study, these instances are referred to as cross-location events. By utilizing a particle filter to keep track of the trajectory of the left and right wrists of each customer, along with the assignment of the RANSAC inlier scores for the tracked keypoints, this problem was addressed. Table 6 illustrates the performance of this system in terms of associating customers and products with cross-location.

Overall, the system was able to achieve an average of 79% correct associations across all runs. Similarly, the deviations in the percent of correct association could be associated with adjacent occlusion. Adjacent occlusion is the occlusion of vision data needed to process and gather pieces of information, such as the location of the keypoints and the bounding box with respect to the shelf. These pieces of information are crucial in the processing of particle filters and RANSAC. Despite this, the percent of correct association per run was at least 60%.

## 8. Real-World Business Use for Case-Enhancing Retail Experience and Optimization

The smart shelf system offers enormous potential for improving customer experience and optimizing operations in brick-and-mortar retail stores.

### 8.1. Inventory Management and Product Placement

The real-time tracking of inventory levels, facilitated by the weight sensors on shelves, detect stock levels and can assist in preventing inventory shortage. This solution reduces the costs associated with manual stock checking labor. Moreover, strategic product placement is enabled by the analysis of customer interaction data, identifying high-traffic areas and providing insights into product appeal. For example, high velocity impulse purchase items can be strategically placed near checkout counters to maximize sales, or slow-moving items could be placed in locations that would improve their velocity.

For the proposed system, the expected replacement policy is that a member of the supermarket staff is responsible for handling item replacements requested by the customer.

Furthermore, to optimize product placement, the ability of the system to detect pickups and putbacks at specific locations of the smart shelf enables the identification of hotspots where customers frequently interact with the shelf. These hotspots can be helpful in formulating data-driven product selection and placement strategies leading to product purchases.

### 8.2. Operational Efficiency

The integration of the smart shelf system with existing retail management systems, such as point-of-sale and inventory management, could facilitate the improvement of data-driven decision making across various business operations [24]. Sale trends and interaction data could be used to optimize purchase decisions and supply chain activities, thereby contributing to more efficient operational processes.

Hence, the ability of the smart shelf to track the frequency of pickups and putbacks gives retailers an insight into customer patterns. For example, the data on the frequency of pickups and putbacks can give the retailer an insight into which products are frequently picked up and returned. Then, from these patterns, they can deduce plans or strategies that could help facilitate a higher probability of purchase for products.

Moreover, the smart shelf system enables retailers to undertake a data-driven approach in optimizing their product placement and strategy. By continuously monitoring pickups and putbacks, the retailers are equipped with a closed-loop cycle of formulated strategies and can execute a new strategy, evaluate against conventional placements, and re-formulate strategies.

### 8.3. Addressing Challenges and Considerations

While the potential of such smart retail technologies is promising, careful consideration must be given to customer privacy and the perception of surveillance [25]. Establishing clear policies and maintaining transparency with customers about data collection and usage is critical. In addition, current brick-and-mortar retail businesses could face technical challenges in the adoption of these AI solutions.

## 9. Conclusions

Weight sensor and computer vision systems were successfully integrated as a smart shelf system. Through data processing and sensor fusion, weight change events detected through the weight system could be associated with a specific customer to help characterize touch-based customer behavior and generate a log of customer–product interactions. The use of a network time protocol server allowed timestamps of both systems to be matched, which then enabled the integration of their data. The weight system performed well with an average of 98% correct event detections and a maximum of 0.01725% invalid readings in a single run. It was also able to detect events that happened in quick succession of each other. The vision system had particularly good detection rates when handling up to three customers at a time. Customers were reidentified by the system after short periods of occlusion. The particle filter and RANSAC implemented in the integration system were able to associate the weight change events with each customer, with a recall rate of 68% or higher. Furthermore, the problem of cross-location was resolved, which can be exemplified by the 79% overall recall rate of the system with cross-location.

Despite the overall success, certain challenges were noted. These include customer stalling and larger groups on the vision data leading to relatively low detection percentages for specific customers. Recommendations for improvement include considering alternative materials and components for shelf and prototype construction, optimizing load cell configurations for weight distribution, and refining wiring through proper PCB implementation. Prospects involve scaling up the system to accommodate additional shelves, exploring the use of multiple cameras in terms of reliability and scalability, and investigating real-time data processing for enhanced system responsiveness. Furthermore, future studies could take the opportunity to fine tune the PACK-RMPF system so that it can take advantage of tracking both the bounding box and the hands, especially in scenarios where multiple people interact with the shelf. Moreover, future experimentation involving multiple product pickups and putbacks is also recommended. Use cases of the smart shelf system could be deployed in small-scale grocery stores.

## Figures and Tables

**Figure 1 sensors-24-00367-f001:**
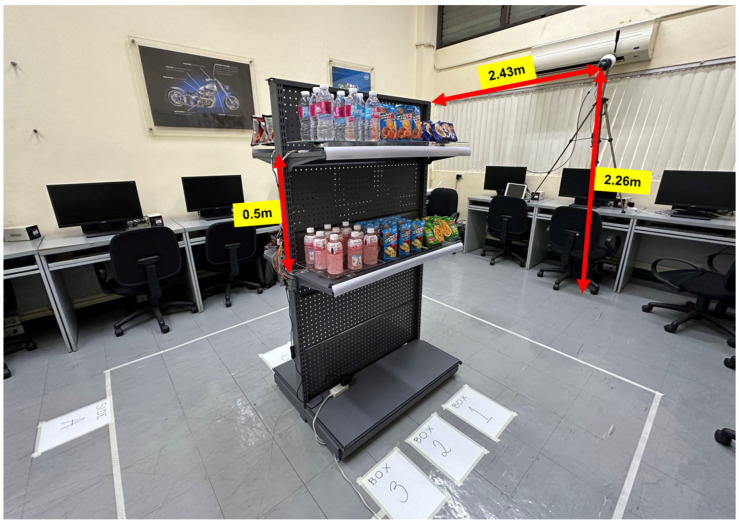
Physical system setup.

**Figure 2 sensors-24-00367-f002:**
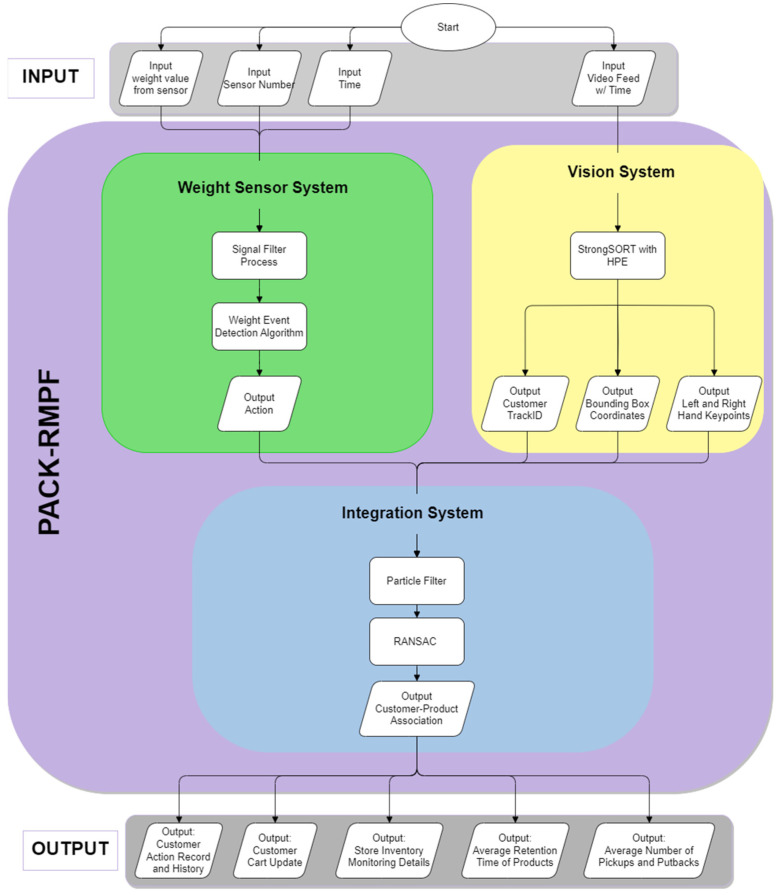
System overview.

**Figure 3 sensors-24-00367-f003:**
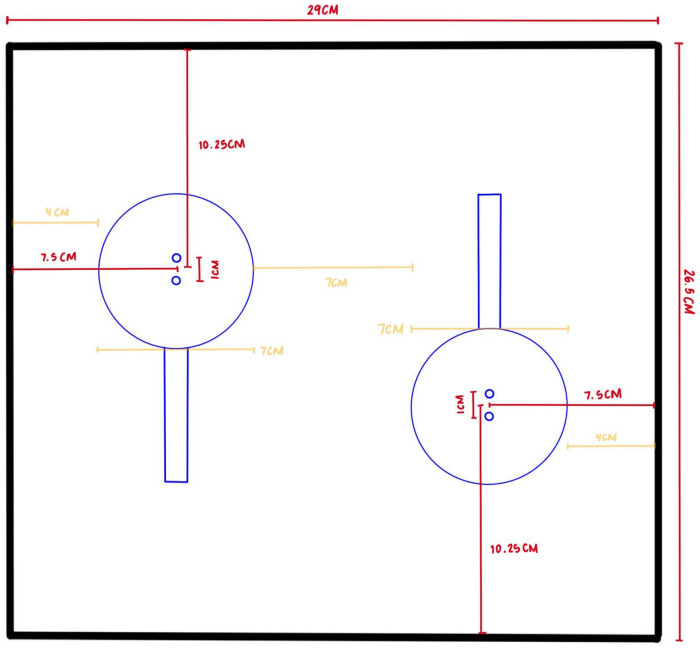
Dimension of each weight bin.

**Figure 4 sensors-24-00367-f004:**
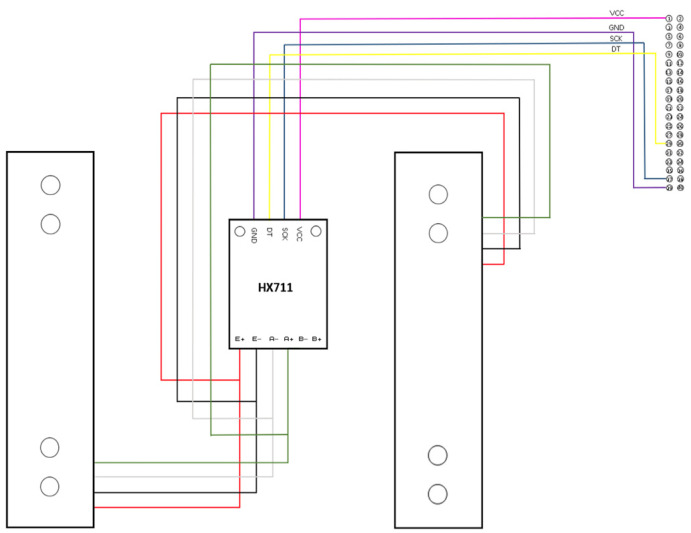
Circuit connection of a weight bin.

**Figure 5 sensors-24-00367-f005:**
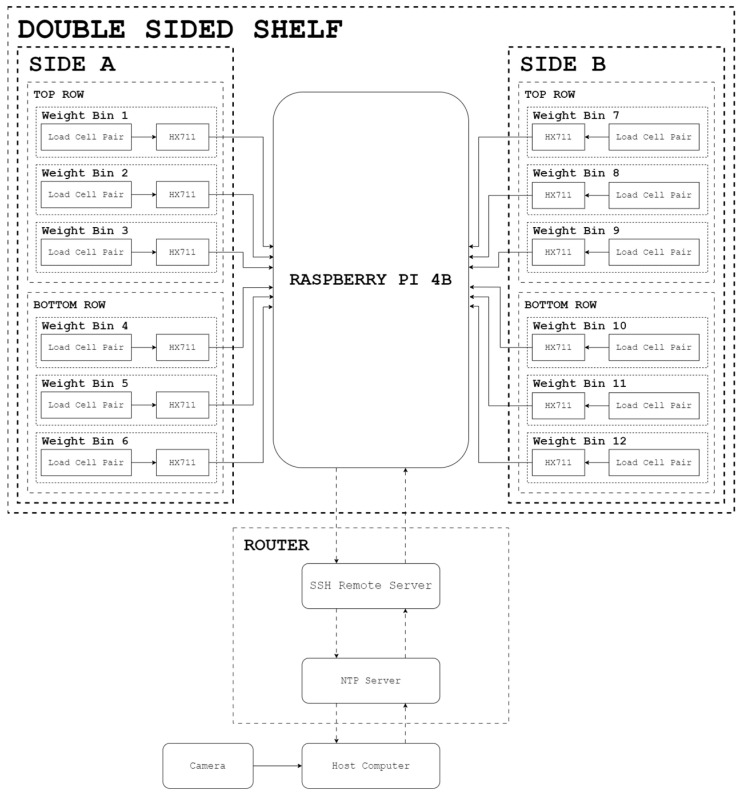
Hardware diagram of the weight sensor array.

**Figure 6 sensors-24-00367-f006:**
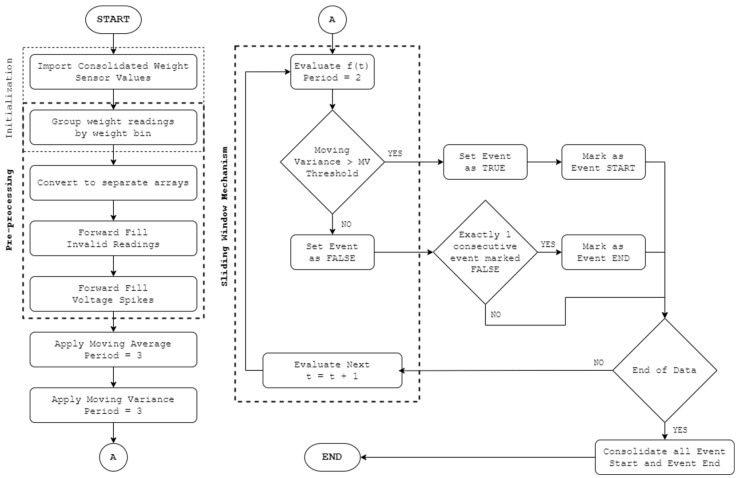
Weight event detection system.

**Figure 7 sensors-24-00367-f007:**
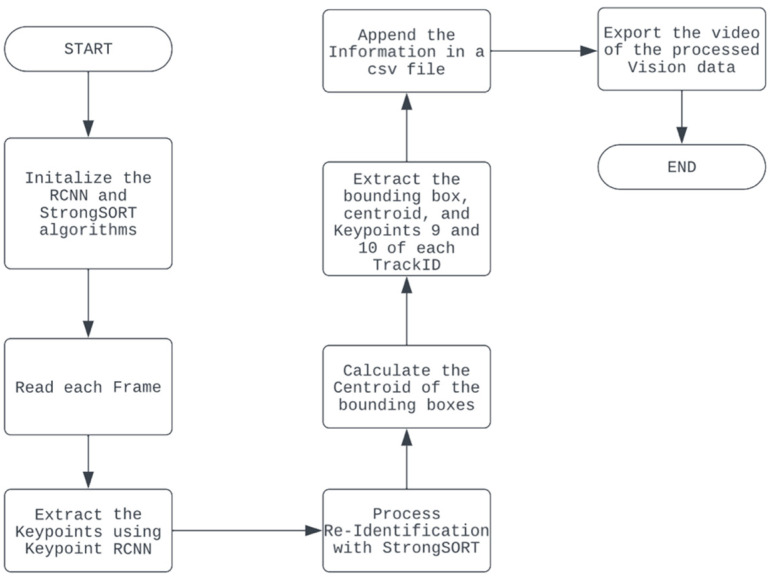
Summary of vision system.

**Figure 8 sensors-24-00367-f008:**
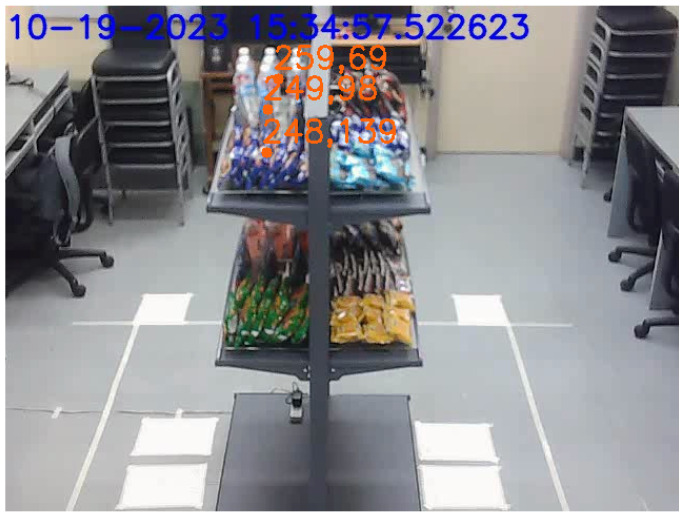
Sample of weight bin landmarks.

**Figure 9 sensors-24-00367-f009:**
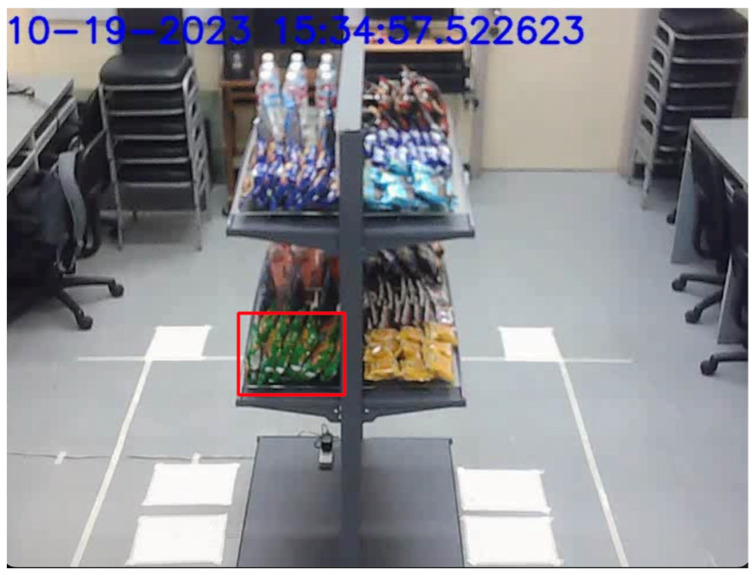
Sample of weight bin pre-defined rectangular area.

**Figure 10 sensors-24-00367-f010:**
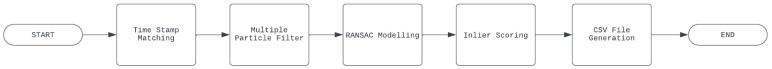
Summary of the PACK-RMPF algorithm.

**Figure 11 sensors-24-00367-f011:**
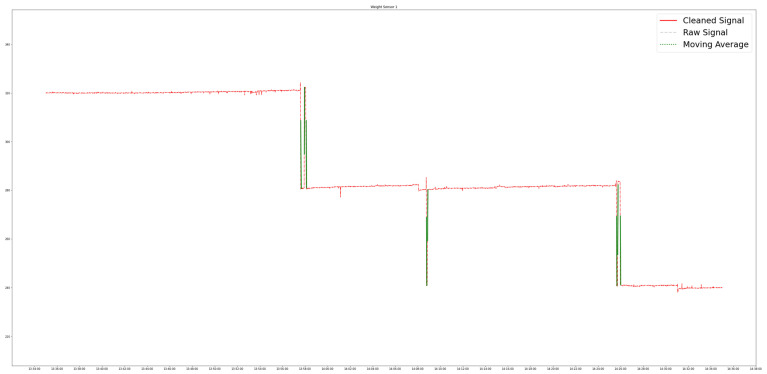
Weight event detection with quick pick ups and putbacks.

**Figure 12 sensors-24-00367-f012:**
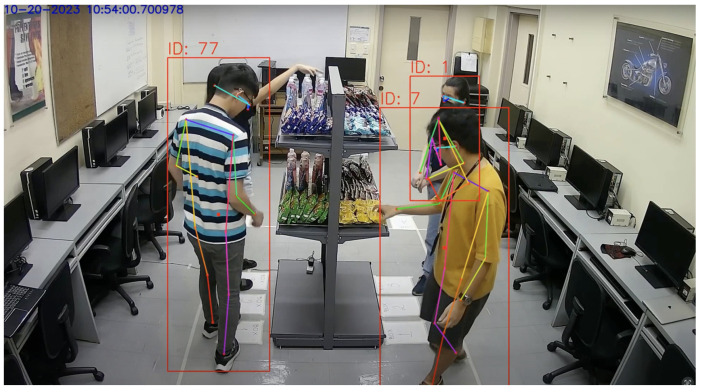
Three people tracked while interacting with the shelf.

**Figure 13 sensors-24-00367-f013:**
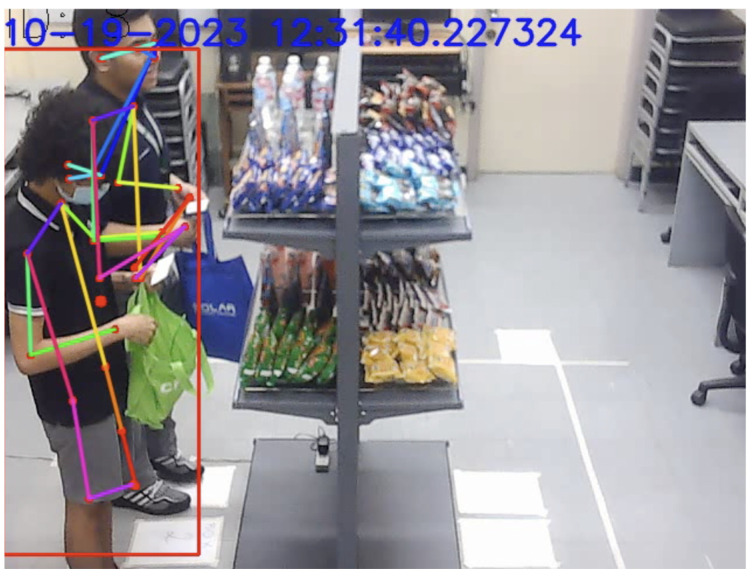
Example of adjacent occlusion.

**Figure 14 sensors-24-00367-f014:**
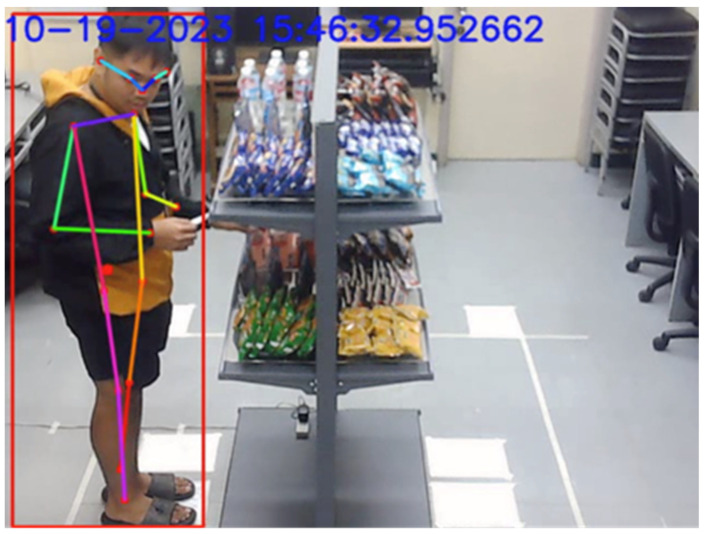
Example of cross-location pick up.

**Table 1 sensors-24-00367-t001:** Moving variance threshold for each weight sensor.

Product	Weight Bin	Weight (g)	$\mathrm{Threshold},\;T_{p}^{\lambda }$
Piattos	HX1	40	100
Cream-O	HX2	33	100
Whattatops	HX3	35	100
Loaded	HX4	32	100
Bingo	HX5	28	50
Cheesecake	HX6	42	7000
Water	HX7	360	1000
Zesto (Orange)	HX8	200	200
Lucky Me (Beef)	HX9	55	200
Mogu Mogu	HX10	330	8000
Zesto (Apple)	HX11	200	200
Lucky Me (Calamansi)	HX12	80	200

**Table 2 sensors-24-00367-t002:** Overall performance report of each weight bin.

Weight Bin	Ratio	False Readings	Total Count
HX1	0.00%	0	9836
HX2	0.04%	4	9875
HX3	0.01%	1	9881
HX4	0.02%	2	9797
HX5	0.02%	2	9828
HX6	0.04%	4	9836
HX7	0.10%	10	9753
HX8	0.03%	3	9759
HX9	0.03%	3	9839
HX10	0.03%	3	9846
HX11	0.01%	1	9835
HX12	0.01%	1	9855
Average	0.03%	2.833333333	9828.333333

**Table 3 sensors-24-00367-t003:** Average percent of associations per run of the bounding box approach of PACK-RMPF.

Run	Duration	Average Percent of Correct Association
1	30 min	74.76%
2	30 min	72.79%
3	30 min	75.77%
4	30 min	91.67%
5	60 min	78.36%
Overall Average Percent of Association		78.67%

**Table 4 sensors-24-00367-t004:** Average percent of associations per run of the hands approach of PACK-RMPF.

Run	Duration	Average Percent of Correct Association
1	30 min	78.36%
2	30 min	81.97%
3	30 min	66.92%
4	30 min	87.77%
5	60 min	75.25%
Overall Average of Percent Association		78.05%

**Table 5 sensors-24-00367-t005:** Average percent of associations per run of PACK-RMPF.

Run	Duration	Average Percent of Correct Association
1	30 min	74.22%
2	30 min	77.55%
3	30 min	68.17%
4	30 min	86.11%
5	60 min	75.62%
Overall Average Percent of Association		76.33%

**Table 6 sensors-24-00367-t006:** PACK-RMPF cross-location results.

Run	Duration	Number of Items with Cross-Location	Number of Items Correctly Identified	Percent of Correct Association with Cross-Location
1	30 min	5	5	100%
2	30 min	2	2	100%
3	30 min	5	3	60%
4	30 min	4	3	75%
5	60 min	5	3	60%
Overall Average Percent of Association				79%

## Data Availability

Data are contained within the article.

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
