# Peer review of "Smart Shelf System for Customer Behavior Tracking in Supermarkets"

_sensors, 2024, doi:10.3390/s24020367_

Round 1

Reviewer 1 Report

Comments and Suggestions for Authors

1.      Introduction: It is recommended to include a literature review in the introduction to highlight the novelty and distinctiveness of this study. The authors can provide a detailed overview of existing smart shelf systems and their limitations and challenges in tracking customer behavior.

2.      Methodology: More technical details and algorithmic explanations regarding the proposed "Product Association with Customer Keypoints through RANSAC Modeling and Particle Filtering (PACK-RMPF)" system are needed. The authors could provide a comprehensive description of the implementation steps for RANSAC and MPF algorithms and explain the rationale behind choosing these algorithms to address the issue of cross-location instances.

3.      Results and Discussion: The paper lacks a section for presenting experimental results and discussing them. This is crucial for evaluating the effectiveness and performance of the proposed method. The authors should provide detailed information about the experimental setup and dataset, showcasing the system's performance under different scenarios. Additionally, a thorough analysis and discussion of the results should be included, highlighting the system's strengths, limitations, and comparisons with existing methods.

4.      Conclusion: It is necessary to add a conclusion section to summarize the main findings and contributions of the paper. The authors can emphasize the potential applications of the proposed system and suggest future directions for improvement.

5.      Citations and References: In the final version, the authors need to ensure the accuracy and completeness of citations and references. It is advised to appropriately cite relevant literature following the journal's citation format and include a comprehensive list of all cited references in the reference list.

Comments on the Quality of English Language

This manuscript require a careful proofreading using reputed online language editing services.

Reviewer 2 Report

Comments and Suggestions for Authors

The paper proposed a framework based on PACK-RMPF, utilizes RANSAC modeling and Multiple Particle Filtering to optimize product placement and store layouts based on detailed customer interactions. Although the proposed framework has many applications in retail industry, however, I have following comments the authors need to consider.

1.       The contribution of proposed work is not clearly mentioned. Please mention what are the research gaps and how proposed method filled those gaps.

2.       The abstract is too long. Please make it precise and concise

3.       Did the author extract features from the raw data or directly provide the data as input to the network?

4.       How the framework extract trajectories and how these trajectories are analyzed to predict the behavior?

5.       How do you perform cleaning of data? As in my opinion, the data must be very noisy. Please provide details

6.       Did you perform any data augmentation as the current size of data is limited to train a deep model?

7.       The proposed framework is pretty much related with action recognition problem. Please discuss the following related reference in the revised manuscript.

a.       Attention-based LSTM network for action recognition in sports. Electronic Imaging

8.       Please discuss the loss function. Also provide details of training the network

9.       Please include a detailed figure of the framework that specifies the inputs and outputs of the framework. The current figures do not portray useful information.

1  The details of the dataset are missing?

1 The experiment is performed in a controlled environment. The authors are requested to include experiments on real time examples.

1Experiment section is weak. Please perform ablation study

1   Discuss failure cases and provide their justifications

Comments on the Quality of English Language

Moderate english editing is required

Reviewer 3 Report

Comments and Suggestions for Authors

Sensors-2766103

The relevance of customer-product contact on shelves is discussed and worked through in this article. In addition, the way goods are arranged on the shelves and how purchases and stocks of goods are delivered to the supermarket owner. The details about the positioning of Camera's is well explained.

My queries raise some valid points,

1. An inadequate explanation is given of particle filtering and its applicability to the suggested methodology.

2. Limited information about the replacement policy of commodities by the customers themselves is found.

3.What positive impact could these smart shelfs have on super markets has to be conveyed.

4.Can you explain the process of collecting and analyzing customer behavior data using the smart shelf system?

5. What types of customer behavior data does the smart shelf system collect and analyze?

6.How is the collected data used to derive insights into customer preferences and purchasing patterns?

7.Can you provide examples of how the system has helped optimize product placement or promotions based on customer behavior?

8.In what ways does the smart shelf system contribute to a personalized shopping experience for customers?

9.How are retailers using the data from the smart shelf system to tailor marketing strategies for individual customers?

10.Have there been noticeable improvements in customer satisfaction since the implementation of the smart shelf system?

11.How does the smart shelf system impact inventory management in the supermarket?

12.Have there been notable changes in stock levels, and how does the system help prevent out-of-stock or overstock situations?

13.In what ways does the system contribute to the overall optimization of the supply chain?

14.What measures are in place to ensure the security of customer data collected by the smart shelf system?

15.How is customer privacy addressed in the implementation of the system?

16.What steps are taken to comply with data protection regulations and build trust with customers regarding their privacy?

17.What challenges have you encountered in implementing and maintaining the smart shelf system?

18.How do you address concerns related to the potential misuse of customer data?

19.Are there any limitations or considerations that retailers should be aware of when adopting such a system?

Comments on the Quality of English Language

To be more accessible, English needs to be upgraded.

Round 2

Reviewer 2 Report

Comments and Suggestions for Authors

Thanks for considering and addressing my concerns

Comments on the Quality of English Language

Moderate english editing are required